# OsABF1 Represses Gibberellin Biosynthesis to Regulate Plant Height and Seed Germination in Rice (*Oryza sativa* L.)

**DOI:** 10.3390/ijms222212220

**Published:** 2021-11-12

**Authors:** Liqun Tang, Huayu Xu, Yifeng Wang, Huimei Wang, Zhiyong Li, Xixi Liu, Yazhou Shu, Guan Li, Wanning Liu, Jiezheng Ying, Xiaohong Tong, Jialing Yao, Wenfei Xiao, Shaoqing Tang, Shen Ni, Jian Zhang

**Affiliations:** 1State Key Lab of Rice Biology, China National Rice Research Institute, Hangzhou 311400, China; liquntang2013@126.com (L.T.); xuhuayu202107@163.com (H.X.); wangyifeng@caas.cn (Y.W.); wanghuimei@caas.cn (H.W.); lzhy1418@163.com (Z.L.); 18338690086@163.com (X.L.); mm123456m@126.com (Y.S.); liguan@caas.cn (G.L.); dearliuwanning@126.com (W.L.); yingjiezheng@caas.cn (J.Y.); tongxiaohong@caas.cn (X.T.); nishen@caas.cn (S.N.); 2College of Life Science and Technology, Huazhong Agricultural University, Wuhan 430070, China; yaojlmy@mail.hzau.edu.cn; 3Institute of Biotechnology, Hangzhou Academy of Agriculture Sciences, Hangzhou 310024, China; xiao_wenfei@126.com

**Keywords:** rice (*Oryza sativa* L.), gibberellins, semi-dwarf, seed germination, polycomb repression complex, bZIP

## Abstract

Gibberellins (GAs) are diterpenoid phytohormones regulating various aspects of plant growth and development, such as internode elongation and seed germination. Although the GA biosynthesis pathways have been identified, the transcriptional regulatory network of GA homeostasis still remains elusive. Here, we report the functional characterization of a GA-inducible *OsABF1* in GA biosynthesis underpinning plant height and seed germination. Overexpression of *OsABF1* produced a typical GA-deficient phenotype with semi-dwarf and retarded seed germination. Meanwhile, the phenotypes could be rescued by exogenous GA_3_, suggesting that *OsABF1* is a key regulator of GA homeostasis. OsABF1 could directly suppress the transcription of green revolution gene *SD1*, thus reducing the endogenous GA level in rice. Moreover, OsABF1 interacts with and transcriptionally antagonizes to the polycomb repression complex component OsEMF2b, whose mutant showed as similar but more severe phenotype to *OsABF1* overexpression lines. It is suggested that OsABF1 recruits RRC2-mediated H3K27me3 deposition on the *SD1* promoter, thus epigenetically silencing *SD1* to maintain the GA homeostasis for growth and seed germination. These findings shed new insight into the functions of *OsABF1* and regulatory mechanism underlying GA homeostasis in rice.

## 1. Introduction

Increasing crop production has been an eternal theme of the world. Previous findings implied that gibberellins (GAs) are major regulators for crop growth and yield through optimizing the harvest index and lodging resistance [1]. In addition, the dormancy and germination process of plant seeds are closely related to GA content, which is a key development stage in crop life cycle and an important factor determining grain yield and quality [2]. As a class of diterpenoids hormones, GAs participate in nearly all important plant growth and developmental events, such as stem and hypocotyl elongation, leaf expansion, seed dormancy and flowering [3,4]. According to the number of carbon atoms in the GA molecule, GAs may be divided into two types: Carbon-20 GAs and Carbon-19 GAs [5]. Thus far, GA biosynthesis pathways have been well documented and have become fairly straightforward. Firstly, in the *ent*-kaurene (the first committed intermediate in GA biosynthesis) biosynthesis progress, geranylgeranyl pyrophosphate (GGPP) is converted into *ent*-copalyl diphosphate synthase (CPS) and *ent*-kaurene synthase (KS) [6]. Secondly, after several oxidation reactions, *ent*-kaurene produces GA_12_-aldehyde [7]. Finally, GA_12_-aldehyde is converted into different GA intermediates and bioactive GAs by GA20-oxidase (GA20ox) and GA3-oxidase (GA3ox), respectively.

Defects of the GA biosynthesis pathways genes usually lead to typical GA-deficient phenotypes such as dwarf, weak germination vigor, and late flowering [8,9,10,11]. One of the most famous examples is *SD1* (*semi-dwarf 1*) encoding an oxidase enzyme in GA biosynthesis [12]. Mutations in the *SD1* reduced the plant height, which conferred rice with lodging resistance and higher harvest index, and finally led to the “green revolution” in rice breeding [12]. It has been known that *GA20ox* converts C-20 from a methyl group to an aldehyde by sequential oxidations and then removes a carbon atom to produce C_19_-GAs (GA_9_ and GA_20_) [13]. *GA20ox* genes have been identified in numerous model plant species, including Arabidopsis [14,15,16] and rice [17,18]. Two types of *GA20ox* have been found: *OsGA20ox1* and *OsGA20ox2* [12,19]. Except for the “Green Revolution Gene”, another *GA20ox* gene, *OsGA20ox1*, also has been identified with similar functions to that of the *OsGA20ox2*. Overexpression of *OsGA20ox1* leads to a taller plant statue, which is a typical GA-overproduction phenotype [20]. Other than the GA biosynthesis genes, large numbers of transcription factors participate in the GAs biosynthesis. For example, *GA20ox1*, which determines the optimal elongation of petiole and hypocotyl under high temperature, can be used as the target gene of transcription factor TCP15 in Arabidopsis thaliana [21], while MADS-box transcription factor SHORT VEGETATIVE PHASE (SVP) inhibits flowering through directly repressing transcription of *AtGA20ox2* [22]. Similarly, Tobacco *RSG* (*REPRESSION OF SHOOT GROWTH*) encoding a bZIP (Basic leucine zipper) transcription activator also controls GA biosynthetic enzyme NtGA20ox1, underpinning cell elongation and shoot growth [23,24].

bZIP transcription factors, as highly conserved types of protein, are widely distributed in eukaryotes. There are about 110 and 89 bZIP transcription factors in Arabidopsis and rice, respectively [25,26]. The differences in structural feature divide bZIPs into 10 subfamilies with diverse functions [26,27,28]. For example, group A is mainly involved in abscisic acid (ABA) or stress signaling [29,30]. As transcription factors, bZIPs can directly bind to the conserved G-box cis elements (CACGTG) in the promoter to activate or repress the transcription of target genes, which underpins various biological processes, including GA biosynthesis and signaling [30,31,32,33,34]. For example, *OsbZIP48* is a homolog of Arabidopsis *AtHY5* controlling photomorphogenesis. Overexpression of *OsbZIP48* in rice reduced the plant height seriously. Further research revealed that *OsbZIP48* regulates GA homeostasis by directly binding to the promoter of *OsKO2* that encodes an ent-kaurene oxidase 2 of the gibberellin biosynthesis pathway [35]. Several other bZIPs such as *ABI3*, *ABI4*, and *ABI5* may also directly or indirectly affect GA accumulation or signaling through ABA signaling pathways [35,36,37,38,39,40,41]. Previous research revealed that *OsABF1* (*LOC_Os01g64730*), an ABA responsive element binding factor, represses rice flowering through indirectly suppressing the transcription of *Early heading date 1* (*Ehd1*) [42]. *OsABF1* was also found to be involved in numbers of abiotic and biotic stress responses, including salt stress, drought stress, and ABA responding, which are very important to plant growth and development [42,43,44]. In this study, we explored the functional characterization of *OsABF1* in GA biosynthesis underpinning plant height and seed germination. *OsABF1* could directly suppress the transcription of green revolution gene *SD1*, thus determining the endogenous GA level in rice. Overexpression of *OsABF1* in rice produced semi-dwarf phenotype with retarded seed germination, which could be restored by exogenous GA. Moreover, OsABF1 interacts with OsEMF2b to recruit PRC2-mediated H3K27me3 deposition on the *SD1* promoter, thus epigenetically silencing *SD1* to maintain the GA homeostasis for growth and seed germination. The rice germplasms with overexpression of *OsABF1*, which caused semi-dwarf phenotype and with retarded seed germination, could be used in rice breeding with lodging resistance and pre-harvest sprouting resistance.

## 2. Results

### 2.1. OsABF1 Overexpression Lines Exhibited GA-Deficient Phenotypes

Two representative overexpression lines, *OxABF1-6* and *OxABF1-9*, with over 300 folds upregulation were used for phenotypical characterization (Appendix A). Compared with wild-type (WT), the *OxABF1-6* and *OxABF1-9* lines showed obvious dwarfism, which was a typical GA-deficient phenotype (Figure 1a). By comparing the length of the internodes of *OxABF1-6**, OxABF1-9*, and WT, we found that all the internodes of *OsABF1* overexpression lines were shortened by 20% to 30%, which finally made the height of *OsABF1* overexpression lines only 73% of that of the WT (Figure 1b,c). To figure out the reason for the semi-dwarf phenotype of *OxABF1-6* plants, we employed paraffin sections to observe the cell size in the second internodes of the *OxABF1-6* and the wild-type (Figure 1d,g). The cell length and width of *OxABF1-6* was significantly decreased as compared with the WT (Figure 1i,j). Moreover, the number of large vascular bundles (LVB) and small vascular bundles (SVB) were also reduced in *OxABF1-6* (Figure 1e,f), indicating that the dwarfism of *OxABF1-6* could be attributed to the reduced cell size and proliferation (Figure 1h). Additionally, we performed seed germination assays on *OsABF1* overexpression seeds and WT. *OsABF1* overexpression lines showed obviously lower germination rate and post-germination growth than the WT (Figure 2a). More importantly, exogenous GA_3_ fully rescued the *OxABF1-6* and *OxABF1-9* germination rates to the WT level, and also partially recovered its post-germination growth (Figure 2c,d,f,g). Meanwhile, application of paclobutrazol (PAC, an endogenous GA biosynthesis inhibitor) significantly inhibited the seed germination rate and post-germination growth of WT seeds to the *OxABF1-6* level but showed no significant effects on those of the *OxABF1-6* (Figure 2b,f,g). Furthermore, hormone quantification assay confirmed that the GA_3_ in 15 days seedlings of *OxABF1-6* reduced over 50% compared with the WT (Figure 2e). Meanwhile, the other endogenous GAs were also quantified, and the results showed that the level of GA_5_, GA_8_, GA_13_, GA_20_, GA_24_, and GA_34_ were significantly decreased in *OxABF1-6* (Appendix A). Thus, we concluded that the observed retarded seed germination and seedling growth of *OxABF1* lines were due to GA deficiency.

Employing the CRISPR/Cas9 technique, we successfully knocked out *OsABF1* and simultaneously knocked out *OsABF1* and its closest homolog *bZIP40* (*LOC_Os05g36160*). The obtained mutants harbored different types of insertions or deletions in the coding sequence of the targeted genes, some of which shifted the open reading frames and reduced the transcriptional levels (Appendix A). However, both *abf1* and abf1 *bzip40* mutants showed no substantial difference with the WT in plant height and seed germination, suggesting that *OsABF1* might be functionally redundant with *bZIP40* and other *bZIP* sibling genes (Figure 1c,d and Appendix A).

### 2.2. OsABF1 Directly Repressed the Transcription of SD1

The GA-deficient phenotype of *OsABF1* overexpression lines suggested that *OsABF1* participated in the regulation of GA biosynthesis or catabolism pathways. Indeed, qRT-PCR analysis revealed that the transcription level of *OsABF1* increased threefold at 1 HAT (hours after GA treatment) and then decreased rapidly at 3 HAT in young seedlings (Figure 3a). Subsequently, the expression levels of several GA-related genes were checked in *OxABF1-6* and the WT by qRT-PCR. As a result, *KS1* (*LOC_Os04g52230*), *KOS1* (*LOC_Os06g37300*), *KOS2* (*LOC_Os06g37224*), *GA2ox3* (*LOC_Os01g55240*), and *GA2ox5* (*LOC_Os07g01340*) were upregulated in *OxABF1-6*, whereas *SD1* (*GA20ox2*, *LOC_Os01g66100*) and *GA2ox1* (*LOC_Os05g06670*) were downregulated in *OxABF1-6* (Figure 3b).

As a crucial gene in GA biosynthesis and the green revolution in rice breeding, *SD1* attracted our particular attention due to its downregulation pattern and the existence of a conserved bZIP binding G-box cis element in its promoter [45] (Figure 3c). In order to check whether *SD1* is a direct target of OsABF1, we performed yeast one-hybrid (Y1H) experiment. In the yeast, OsABF1 successfully activated the expression of *proSD1::LacZ* reporter, thus OsABF1 may directly bind to the *SD1* promoter (Figure 3e). Moreover, we conducted electrophoresis mobility shift assay (EMSA) to test the binding in vitro. The GST-OsABF1 protein could physically bind to the *SD1a* probe containing the G-box, and the excess unlabeled competitive probes effectively alleviated the biotin probe bound by GST-ABF1 (Figure 3d). ChIP-qPCR result further demonstrated that OsABF1 was significantly enriched in the regions P3 of *SD1* promoter where the G-box cis element was located (Figure 3f). Finally, we performed luciferase (LUC) transient transcriptional activity assay to detect the effect of OsABF1 on *SD1* transcription in rice protoplasts. The result showed that the luciferase activity of the *proSD1::LUC* reporter was significantly repressed when OsABF1 was introduced. The result indicated that *OsABF1* acted as a transcription repressor on *SD1*, which consistent with the reduced *SD1* expression in *O**sABF1* overexpression lines (Figure 3g,h). Taken together, we proposed that OsABF1 could directly bind to the *SD1* promoter and repress its transcription to regulate GA biosynthesis.

### 2.3. OsABF1 Bound to OsEMF2b, Causing the Deposition of H3K27me3 on the SD1 Promoter

Polycomb group (PcG) proteins have been shown to regulate growth and development in plants via mediating H3K27me3-suppressing histone modification on the target genes, which has been recognized as a significant epigenetic mechanism in gene regulation [46,47]. The PcG proteins in rice include two E(z)-like homologues (OsCLF and OsiEZ1), two Su(z)12 homologues (OsEMF2a and OsEMF2b), and two ESC homologues (OsFIE1 and OsFIE2), among which *OsEMF2b* has been reported to control plant height and seed germination by regulating the expression of *OsVP1* [48,49]. As mentioned above, the overexpression of *OsABF1* reduced the plant height and seed germination; therefore, we speculated that OsABF1 may interact with OsEMF2b to recruit PcG-mediated repression on *SD1*. The hypothesis was tested by yeast two-hybrid assays, which showed that OsABF1-BD physically binds to OsEMF2b-AD in yeast (Figure 4a). In the GST pull-down assay, the purified His-OsEMF2b was pulled down with GST-OsABF1, but not with GST tag, indicating that OsABF1 interacts with OsEMF2b in vitro (Figure 4c). The BiFC experiment in plant also revealed that the nYFP-OsEMF2b and OsABF1-cYFP pair produced strong fluorescent signals in the nucleus of tobacco leaf cells, whereas no fluorescence signals were observed when nYFP-OsEMF2b or OsABF1-cYFP was expressed alone (Figure 4b). Therefore, these results strongly demonstrated that OsABF1 binds to OsEMF2b. Finally, we performed ChIP-qPCR to test the deposition patterns of OsEMF2b and its mediated histone modification H3K27me3 on the *SD1* region. The results showed that both OsEMF2b and H3K27me3 were significantly enriched in the *SD1* region, particularly the P3 region, which is largely overlapped with the OsABF1 deposition patterns (Figure 4d,e). Hence, OsABF1 represses the *SD1* transcription via recruiting the PcG-mediated H3K27me3. 

### 2.4. OsEMF2b and OsABF1 Had Antagonistic Roles in GA Biosynthesis Regulation

In contrast to the GA-induced expression pattern of *OsABF1*, the transcription of *OsEMF2b* was significantly downregulated by exogenous GA in young seedlings at 1 HAT, suggesting the antagonistic roles of *OsABF1* and *OsEMF2b* in GA biosynthesis regulation (Figure 3a). We adopted an *OsEMF2b* T-DNA insertional mutant from another lab for the characterization of its biological function [47]. The *emf2b* contains a T-DNA insertion in the fourth exon, which largely disrupted the expression of the gene (Appendix A). The homozygous *emf2b* mutants displayed a similar but more severe dwarf phenotype than *OsABF1* overexpression lines (Figure 5a–c). Unfortunately, we were not able to test the seed germination of *emf2b* mutants due to the complete sterility of the homozygous *emf2b* lines (Appendix A). In *emf2b*, *OsABF1* was significantly elevated (Figure 5d). Meanwhile, *OsEMF2b* expression was drastically reduced in *OsABF1* overexpression lines (Figure 5e). We also performed qRT-PCR to examine the expression of GA-related genes in *emf2b* and the WT and found that all the tested genes showed the same differential expression pattern in *emf2b* as in *OsABF1* overexpression lines (Figure 5f). In particular, *GA20ox2/SD1* expression in *emf2b* was decreased to only 1% of that in the WT, which well explained the extremely dwarf nature of *emf2b*.

## 3. Discussion

### 3.1. OsABF1 Targets SD1 to Regulate GA Biosynthesis

The growth and development of plants are regulated by many factors in vivo and vitro, including light, temperature, water, and genetic factors. The plant hormones are of the most importance in regulating the growth and development, for which the biosynthetic pathways are adjusted by many transcription factors. In this study, we revealed that a bZIP transcription factor Os*ABF1* regulates GA biosynthesis via targeting the green revolution gene *SD1*. Although the knock-out mutants of *OsABF1* and the double mutants *abf1bzip40* grew as normal as the WT (Appendix A), the overexpression of Os*ABF1* gave rise to typical GA-deficient phenotypes such as semi-dwarf and later seed germination (Figure 1a–c and Figure 2a). Meanwhile, overexpression of Os*ABF1* exhibited significantly lower GA_3_ concentration than the WT (Figure 2e). More importantly, the application of exogenous GA_3_ successfully restored the phenotype of *OsABF1* overexpression lines to the WT level, suggesting that GA deficiency is the major reason for the observed phenotypes (Figure 2c,d). Actually, *OsABF1* has also been reported as a suppressor of rice flowering [42]. Given that GA is a major hormone promoting plant flowering, the later flowering phenotype should be attributed to the GA deficiency in *OsABF1* overexpression lines as well. Moreover, Y1H, EMSA, and LUC transient transactivation and ChIP-qPCR assays confirmed that OsABF1 could directly bind to the G-box cis elements in the promoter regions of *SD1* to repress its transcription, which strongly suggested that *SD1* is a key node of the *OsABF1*-governed GA homeostasis (Figure 3c–h). 

*SD1* has been identified as an oxidase enzyme in GA biosynthesis. Mutations in the *SD1* led to the “green revolution” in rice breeding by reducing the plant height with stronger lodging resistance and higher harvest index [12]. Meanwhile, the *qSD1-2*, another allele of *SD1*, was found involved in the reduction of GA synthesis, seed dormancy enhancement, and plant height reduction [8]. In breeding practice, double-stranded RNAi on two *SD1* fragments influenced the plant height in different extents [50]. Gene editing on *SD1* enabled the rapid improvement of landraces with shorter plant height and enhanced lodging resistance without sacrificing many other excellent agronomic traits such as low phosphorus tolerance and broad-spectrum resistance to a variety of diseases and insect pests [51]. 

### 3.2. OsABF1 May Mediate the Antagonistic Relationship of ABA and GA 

Like indispensable lubrication in plants lifecycle, various plant hormones regulate plant growth and development synergistically or antagonistically, their relationship woven into a series of complex networks in plants [32,52,53,54]. Abscisic acid (ABA) and GA work antagonistically in regulating various plant growth and development events such as seed dormancy and germination, root growth, plant height, flowering initiation, and abiotic stress [11,55,56]. It was found that exogenous ABA usually reduces GA accumulation, while ABA-deficient or ABA-signaling mutants were able to bypass the GA requirement for seed germination [57]. Thus far, a large number of mediators of ABA–GA antagonism have been identified. In Arabidopsis, *ABF4* significantly increased the level of ABA and promoted the transcriptional regulation of GA metabolism genes, which suggested that *ABF4* was a hub of ABA and GA signaling pathway [58]. It was revealed that SnRK2s-APC/CTE regulatory module mediated the antagonism between GA and ABA signal transduction in higher plants [59]. A very recent study from our lab also reported that rice *WRKY72* inhibits GA accumulation and seed germination via the “WRKY72-*LRK1*-*OsKO2*” pathway [60]. Given that WRKY72 is a phosphorylation substrate protein of the ABA signaling core component SAPK10 [61], WRKY72 may serve as a key node in the ABA–GA interaction. Actually, several pieces of evidence have shown that *OsABF1* is functionally implicated in ABA signaling. Transcription of *OsABF1* is significantly induced by ABA and is able to bind to the ABA-responsive cis elements [44]. Its mutants became more sensitive to drought and salinity treatments, which are closely related to the functions of ABA. Additionally, its closest Arabidopsis orthologs, *OsABF1*, *2*, *3*, and *4*, are also ABA-inducible and functionally involved in abiotic stress responses [62]. Combining the roles of *OsABF1* in ABA signaling and GA biosynthesis suppression, it is rational to propose that *OsABF1* is a key player in mediating the antagonistic relationship of ABA and GA.

### 3.3. OsABF1 Interacts with OsEMF2b-PRC2 Complex to Suppress SD1 Transcription through H3K27me3

Higher eukaryotes have evolved profound epigenetic regulatory mechanisms to fine-tune gene transcription. Initially identified in Drosophila, PcG proteins have been reported to inactivate gene expression and maintain the silencing state of the target chromatin through covalent histone modifications [63,64]. PcG proteins contain polycomb repressive complex 1 (PRC1) and PRC2. PRC1 complexes are E3 ubiquitin ligases depositing mono ubiquitylation of lysine 119 mark on histone 2A (H2AK119ub), while PRC2 majorly catalyzes tri-methylation on histone H3 at lysine 27 (H3K27me3) to repress target transcription [48,49,64]. PRC2 is not able to bind to the target inherent DNA, due to lack of DNA binding domains. Thus, its localization to target sites largely relies on other transcription factors that can link PRC2 and the DNA motifs [65]. Recently, recruitment of PRC2 by transcription factors has been demonstrated as a significant mechanism in gene transcriptional regulation. For example, in 2016, two independent labs reported that VAL1 and VAL2 bound to the RY motifs in the *FLC* promoter, wherein they recruit PRC2-mediated H3K27me3 for FLC repression and flowering promotion after vernalization [66,67]. Similarly, OsLF recruits PRC2 to the CANNTG (bHLH cis-element) motif of *Hd1* by interacting with PRC2 component OsVIL2 and represses the *Hd1* expression via H3K27 tri-methylation modification [68]. In the current study, we demonstrated that OsABF1 recruits EMF2b-mediated H3K27me3 on the promoter of *SD1*, thus epigenetically regulating GA biosynthesis. This conclusion has been supported by several lines of evidence including (1) OsABF1 being a direct repressor of *SD1* (Figure 3c–e); (2) OsABF1 being able to physically interact with PRC2 component OsEMF2b, in vitro and vivo (Figure 4a–c); and (3) the enriched regions of OsABF1, OsEMF2b, and H3K27me3 in the *SD1* promoter largely overlapping (Figure 3f and Figure 4d,e). Together with other reports mentioned above, our study suggests that plant growth and development may subject to a very general regulatory mechanism that transcription factors repress target gene transcription via recruiting PRC2-mediated H3K27me3 modification.

In addition, there might be a regulation mechanism that overexpression of *OsABF1* could repress the expression of *SD1*. In the meantime, enrichment of OsEMF2b would mediate more H3K27me3 in the promoter of *SD1*, and also repress its expression. Further, in plants, there must be a regulation mechanism of maintaining the GA homeostasis, namely, the moderate and stable expression of *SD1*. Thus, when *OsABF1* was overexpressed, then the expression of *OsEMF2b* was reduced (Figure 5e), whereas when *OsEMF2b* mutant to *emf2b*, then the expression of *OsABF**1* was significantly increased to repress the expression of *SD1* (Figure 5d) and then maintain the moderate and stable expression and the GA homeostasis in plant. 

## 4. Materials and Methods

### 4.1. Plant Materials and Growth Conditions

The complete coding sequence of *OsABF1* was amplified using the cDNA of Nipponbare (*Oryza sativa* ssp. japonica) leaf and inserted into the fusion vector pCAMBIA1390 driven by the ubiquitin promoter with restriction digestion site of *Kpn*I and *Bam*HI. To produce *OsABF1* mutant lines, we used a reported CRISPR/Cas9 technique to generate *abf1* and *abf1bzip40* double mutants [69]. All the binary vectors were transformed into the WT (*japonica* cv. Nipponbare) by Agrobacterium tumefaciens-mediated method. 

The *emf2b* (A T-DNA insertion mutant of *OsEMF2b*) lines in *japonica* cv. Dongjin were adopted from other labs [50]. To measure plant height and test seed germination rate, we cultivated all plants during the normal rice-growing season in the experimental field of China National Rice Research Institute, Hangzhou (119°57′ E, 30°05′ N). The primers and sgRNA sequences are provided in Appendix A.

### 4.2. Seed Germination and Seedling Growth Assays

The experiment of seed germination was conducted on the basis of the previous study with minor revision [32]. Briefly, all of the dehulled seeds were sterilized in 70% ethanol for 2 min, then in 50% NaClO for 40 min, and washed in sterilized water until clean. Then, the seeds were placed on 1/2 strength Murashige and Skoog medium (pH 5.8) and incubated in a growth cabinet (28 ± 2 °C, 12/12 h photoperiod and 70% relative humidity) for 7 days. As the seed coleoptile reached 5 mm in length, this phenotype was defined as complete germination. Seed germination rate was recorded every 12 h, and three biological replications (each replicate including 50 seeds) were used for each sample. Seedling height was measured after 7 days of germination. For the hormone treatment, different concentrations of GA_3_ and PAC were added into 1/2 strength Murashige and Skoog medium treatment.

### 4.3. Hormone Quantification

For GAs quantification in Nipponbare and *OxABF1-6* plants, 3 g of 15-days old fresh seedlings of Nipponbare and *OxABF1-6* were harvested and then frozen in liquid nitrogen. Quantification of endogenous GAs levels was measured by liquid chromatography–tandem mass spectrometry system. This method was conducted as described in previous studies, with a few modifications [70,71]. All samples were extracted with 1.5 mL methanol with concentration of 80% at 4 °C for 16 h, then centrifuged at 13,000× *g* at 4 °C for 10 min. Further, the supernatant was collected and evaporated to dryness under a nitrogen gas stream. Finally, it was reconstituted in 100 mL of 95% acetonitrile for the liquid chromatography–tandem mass spectrometry analysis. 

### 4.4. Paraffin Sections

Internodes of WT and *OxABF1-6* collected at post-heading stage were soaked in FAA (70% ethanol 90 mL, 5 mL acetic acid, and 37–40% formaldehyde (5 mL)) solution for about 24 h, then dehydrated through gradient ethanol concentrations followed by different graded chloroform concentrations. Finally, all the treated samples were soaked in wax, embedded and sliced according to the previous study [72]. All tissues were stained with Safranine O-Fast Green Stain Kit (YESEN Biotech, Shanghai, China) and sliced into 10-mm sections using a microtome (Leica, Wetzlar, Germany). Images were photographed using a Leica DM2500 microscope (Leica, Wetzlar, Germany). The cell size was measured by ImageJ software (ImageJ V.1.8.0).

### 4.5. Quantitative Reverse Transcription (qRT)-PCR 

RNA isolation kit (YESEN Biotech, Shanghai, China) was used to extract total RNA. cDNAs were synthesized by Prime Script RT Master Mix (Takara, Dalian, China) and quantitative real-time PCR of various genes was conducted by CFX 96 real-time PCR instrument (Bio-Rad, California, USA). Each sample was performed with three technical repetitions, and *UBQ* (*LOC_Os03g13170*) was used as internal control. The transcription levels were calculated by the 2^−^^△△CT^ method. Primers used in this study were listed in Appendix A.

### 4.6. Yeast One-Hybrid Assay

Yeast one-hybrid system (Clontech, CA, USA) was used to check the interaction between OsABF1 and the *SD1* promoter. The complete coding sequence of *OsABF1* was fused to the pB42AD vector with *Eco*RI and *Xho*I sites. The promoter of *SD1* (1500 bp upstream of the transcriptional start site) was amplified and inserted into the pLacZi vector using *Eco*RI and *Kpn*I sites. All fused vectors were introduced into EGY48 (yeast strain), then grown on SD/-Ura/-Trp plates.

### 4.7. Electrophoresis Mobility Shift Assay (EMSA)

The probes of *sd1* were synthesized and labeled with Cy5.5 by SUNYA biotech (Hangzhou, China). Non-labeled probes were used as competitors. For the Cy5.5-labeled probe, all the process of the EMSA was performed under darkness. Odyssey CLx infrared fluorescence imaging system (LI-COR, Lincoln, NE, USA) was used to detect the fluorescence signal of the EMSA gel. Probes used were listed in Appendix A.

### 4.8. Luciferase Transient Transcriptional Activity Assay 

For the luciferase activity assay, 1500bp upstream of the *SD1* transcription start site was amplified and inserted into 190LUC vector with *Hin*dIII and *Bgl*II to generate the reporter. The CDS of *OsABF1* was fused into “None” with *Bam*HI and *Eco*RI as an effector. Both fusion vector plasmids were co-transformed into rice protoplast, and the luciferase activity was analyzed by the Luciferase Reporter Assay System (Promega, Madison, WI, USA). Rice protoplast extraction and transformation were conducted according to a predecessor study [73]. Primers used were listed in Appendix A.

### 4.9. ChIP-qPCR

The ChIP-qPCR assay was conducted following the previously published protocol [48]. OsABF1 and OsEMF2b antibodies were commercially synthesized by Gene script company (Shanghai, China), and H3K27me3 antibody was purchased (Cat No: Ab6002. Abcam, Cambridge, MA, USA). Chromatin was extracted from 3 g of 15 DAG (days after germination) *OsABF1* overexpression seedlings and then cross-linked with 1% formaldehyde under vacuum. Then, the extracted chromatin was fragmented to about 300 bps in size by sonication. Finally, DNA/protein complex was immuno-precipitated with polyclonal OsABF1, OsEMF2b, and H3K27me3 antibodies. Triple biological replicates of each sample were performed, and the data analysis was conducted according to previously reported methods [46,49]. Primers used were listed in Appendix A.

### 4.10. Yeast Two-Hybrid Analysis and Bimolecular Fluorescence Complementation 

The yeast two-hybrid assay was conducted on the basis of the manufacturer’s protocol (Clontech, Palo Alto, CA, USA). The full-length CDS of *OsABF1* and *OsEMF2b* were amplified and separately fused into bait vector pGBKT7 and prey vector pGADT7. pGBKT7-OsABF1 and pGADT7-OsEMF2b were co-transformed into yeast strain Y2H Gold. pGBKT7-53 and pGADT7-T were used as positive controls, pGBKT7 and pGADT7 were used as negative controls. For the bimolecular fluorescence complementation (BiFC) experiment, *OsABF1* CDS without stop codon was fused into pDOE vector at the *Bsp*eI site, and the complete coding sequence of OsEMF2b was inserted into pDOE vector at the *Bam*HI site. Then, the fusional expression vectors were transformed into Agrobacterium strain EHA105 by electroporation.

### 4.11. In Vitro Pull-Down Assay

For the vitro pull-down assay, the full CDS of the *OsABF1* and *OsEMF2b* were amplified and ligated into the pGEX-4T-1 and pET28a vectors, respectively. The fused constructs and empty pGEX-4T-1 were introduced into *E. coli* strain Transetta (Transgen, Beijing, China). The primers were provided in Appendix A.

## Figures and Tables

**Figure 1 ijms-22-12220-f001:**
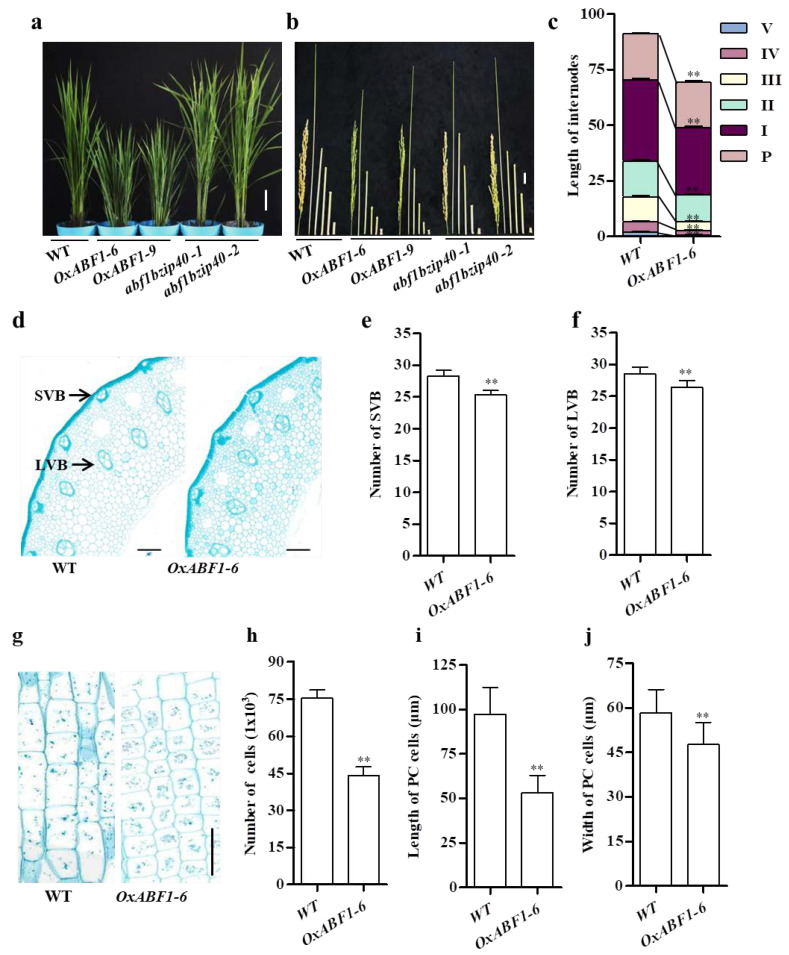
The plant height and length of internodes were reduced in *OsABF1* overexpression plants. (**a**) Plant morphology of *OxABF1-6*, *abf1bzip40-1*, and WT (Nipponbare) at the start heading stage, scale bar = 20 cm. (**b**) The panicle and internodes of *OxABF1-6*, *OxABF1-9*, *abf1bzip40*, and WT (Nipponbare). Bar = 2 cm. (**c**) Statistical analysis of plant height and internode length of *OxABF1-6* and WT (Nipponbare). Data are shown as means ± SD (*n* = 15). (**d**) Cross-section of the second internode in WT and *OxABF1-6*, scale bar = 200 μm. SVB: small vascular bundle; LVB: large vascular bundle. (**e**,**f**) Number of SVB and LVB calculated from cross-sections of the second internode; values are shown as means ± SD (*n* = 20). (**g**) Longitudinal section of the second internode in WT and *OxABF1-6*, scale bar = 200 μm. (**h**) Cell numbers of the second internodes of WT and *OxABF1-6*; values are shown as means ± SD (*n* = 20). (**i**,**j**) The parenchyma cell (PC) length and width. Data is shown as means ± SD (*n* = 100). Throughout this figure, ** *p* < 0.01 by Student’s *t*-test analysis.

**Figure 2 ijms-22-12220-f002:**
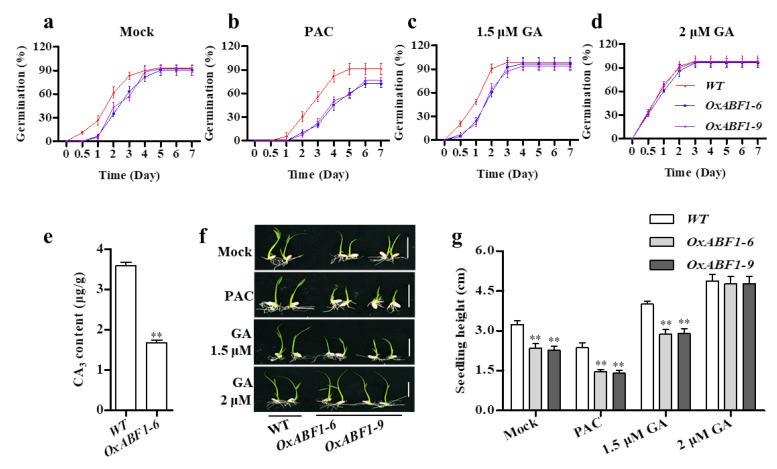
Seed germination characteristics of *OsABF1* overexpression lines. (**a**–**d**) Seed germination time courses of the WT, *OxABF1-6*, and *OxABF1-9* under mock (**a**), 10 µM PAC treatment (**b**), 1.5 µM GA_3_ (**c**), and 2 µM GA_3_ (**d**). (**e**) Quantification of GA derivatives of WT and *OxABF1-6* seedlings analyzed with liquid chromatography–tandem mass spectrometry. Error bars represent means ± SD (*n* = 3), ** indicates statistically significant difference at *p* < 0.01 compared with Nipponbare in Student’s *t*-test. (**f**,**g**) After 7 days seed germination, phenotypes and seedlings height of the WT, *OxABF1–6*, and *OxABF1–9* under mock, 10 µM PAC treatment, 1.5 µM GA_3_, and 2 µM GA_3_. Error bars represent means ± SD (*n* = 20). ** *p* < 0.01 by Student’s *t*-test analysis.

**Figure 3 ijms-22-12220-f003:**
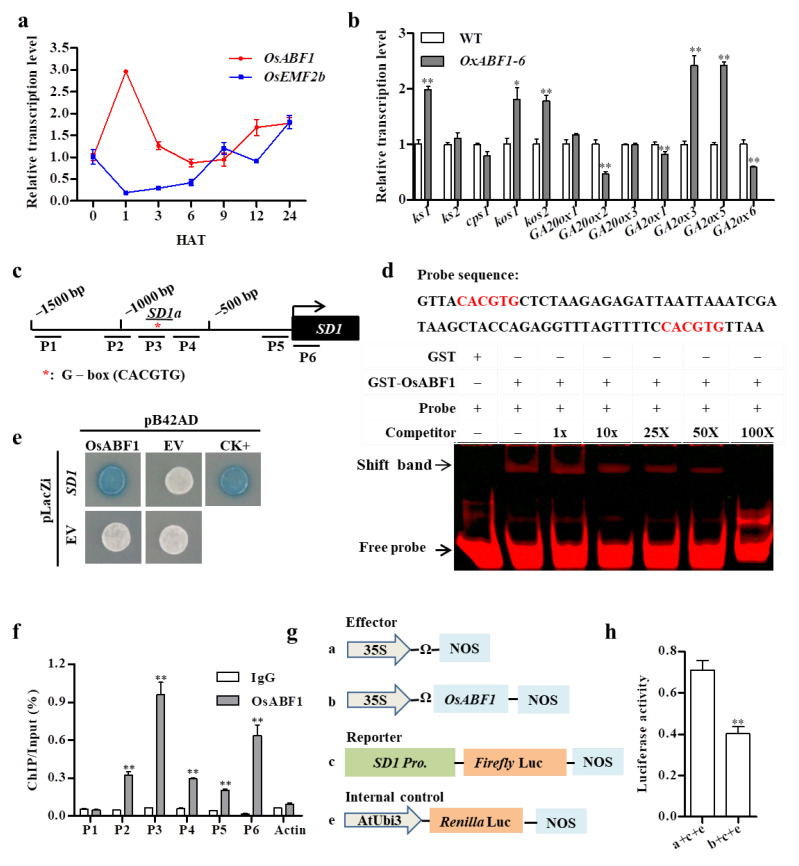
OsABF1 directly bound to the *SD1* promoter and repressed its transcription. (**a**) Time-course expression level of *OsABF1* in 15-day-old seedlings upon 100 μM exogenous GA_3_ treatment. HAT: hours after GA treatment. (**b**) The expression levels of GA-related genes in WT and *OxABF1-6* leaves at tillering stage. (**c**) Schematic presentation of the *SD1* promoter regions (1500bp upstream of the *SD1* transcription start site). Bar 1 to 6 indicate the regions checked by ChIP-qPCR. The red asterisk (*SD1a*) represents the position of the G-box motif. (**d**) EMSA assay showed OsABF1 directly binds to the promoter of *SD1*. The 1×, 10×, 25×, 50×, and 100× fold excess non-labeled probes were used for competition. The red letter represented G-box. (**e**) Yeast one-hybrid assay showed that OsABF1 could bind to the *SD1* promoter (1500 bps upstream of the *SD1* transcription start site); UDT1-pB42AD and RMS2-pLacZi were used as CK^+^. (**f**) ChIP-qPCR analysis of the OsABF1 enrichment on *SD1* promoter in *OxABF1-6*. (**g**,**h**) Luciferase (LUC) transient transcriptional activity assay in rice protoplast. Error bars represent means ± SD (*n* = 3). * *p* < 0.05, ** *p* < 0.01 by Student’s *t*-test analysis.

**Figure 4 ijms-22-12220-f004:**
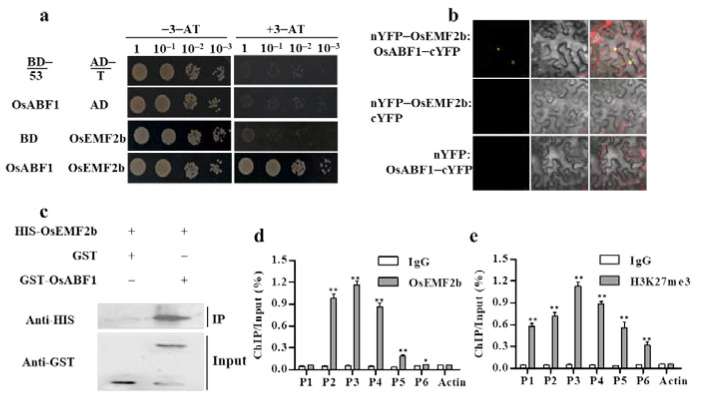
OsABF1 recruits PRC2 complex to target loci by interacting with OsEMF2b. (**a**) Yeast two-hybrid assay showed OsABF1 interaction with OsEMF2b in yeast. Yeast cells co-transformed with OsABF1-BD and OsEMF2b-AD were grown on SD-Trp/-Leu/-Ade/-His selective medium with 3 mM 3-AT (a competitive inhibitor of the HIS3 gene product). (**b**) BiFC was used to detect the interaction between OsABF1 and OsEMF2b. The tested protein pairs were constructed in vector pDOE-BiFC and transiently expressed in tobacco leaf epidermal cells. Positive interactions are indicated by the visualization of YFP fluorescence, scale bar = 20 µm. (**c**) In vitro pull-down assay. Purified His-OsEMF2b and GST-OsABF1 were subjected to pull-down assays and detected with anti-His and anti-GST antibodies, respectively. (**d**,**e**) ChIP-qPCR analysis of OsEMF2b and H3K27me3 deposition on *SD1* target loci in *OxABF1-6*. Values are means ± SD (*n* = 3). * *p* < 0.05, ** *p* < 0.01 by Student’s *t*-test analysis.

**Figure 5 ijms-22-12220-f005:**
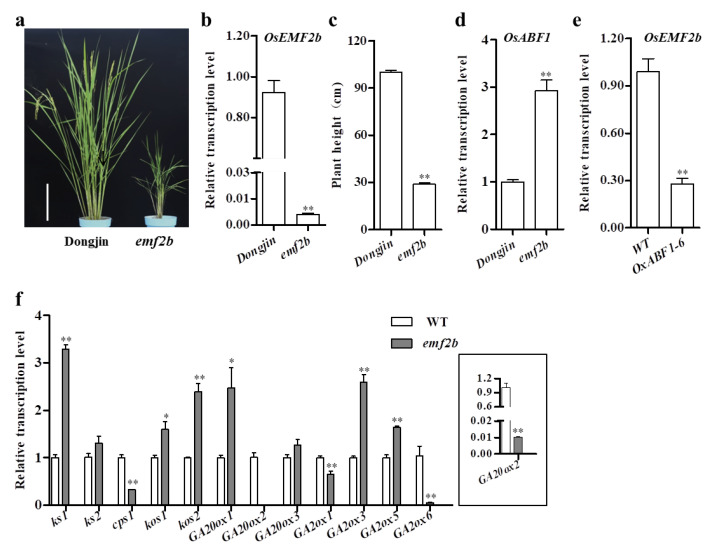
The plant height of *emf2b* was reduced. (**a**) The phenotype of WT (Dongjin) and *emf2b*. (**b**,**d**) *OsEMF2b* and *OsABF1* transcriptional level in WT (Dongjin) and *emf2b* leaves at tillering stage. (**c**) The plant height of WT (Dongjin) and *emf2b.* (**e**) *OsEMF2b* transcriptional level in WT (Nipponbare) and *OxABF1-6* leaves at tillering stage. (**f**) The expression levels of GA-related genes in WT (Dongjin) and *emf2b* leaves at tillering stage. Throughout this figure, * *p* < 0.05, ** *p* < 0.01 by Student’s *t*-test analysis.

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
