# Peer review of "OsABF1 Represses Gibberellin Biosynthesis to Regulate Plant Height and Seed Germination in Rice (Oryza sativa L.)"

_ijms, 2021, doi:10.3390/ijms222212220_

Round 1
Reviewer 1 Report
The manuscript reports functional characterization of OsABF1 in GA biosynthesis underpinning plant height and germination in rice. It is overall well-written and deserves close attention for publication in IJMS. Please consider my comments below to improve the manuscript.
I think the first paragraph in “Introduction” is a bit too long. I suggest breaking it down into 2-3 smaller paragraphs according to different main ideas (for example, one paragraph on GA biosynthesis and another paragraph on rice genes involved in this process).
In the second paragraph in “Introduction”, could you briefly add why it is important to functionally characterize the role of OsABF1 on plant height and seed germination, probably by emphasizing the importance of those traits in agriculture? What about moving line 258-264 somewhere into “Introduction” instead of the “Discussion” section?
In line 69, the full name of bZIP should appear earlier when the term is used first in line 66.
The first sentence in the result section (line 96) should be removed.
In Fig. S1 a and b, the genic locations of the CRISPR/Cas9 induced mutations should be indicated along with their protein-level effects. Which exon is affected by each mutation? Are they all frameshifts?
I am not convinced by line 130-132, which states OsABF1 might be functionally redundant with bZIP40 based on the observation that both abf1 single and abf1bizip40 double mutants showed no phenotypic effects on plant height and germination. Could you elaborate this more?
Please revise the orders of supplemental figures so they appear in the text in alphanumerical order. Currently, Fig. S1c appears first followed by Fig. S2h, and then Fig. S1 a-b. This order is somewhat confusing for readers.
In line 157-160, indicate what “treatment” you are describing about.
In Fig. 3c, the legend mentions that there are 10 bars (line 190) but only six bars are presented. The location of the G-box element mentioned in line 168 should be also included in Fig. 3c.
Author Response
Response to Reviewer 1 Comments
The manuscript reports functional characterization of OsABF1 in GA biosynthesis underpinning plant height and germination in rice. It is overall well-written and deserves close attention for publication in IJMS. Please consider my comments below to improve the manuscript.
Response: Thank you for your appreciation! We fully considered the suggestive comments and made a point-to-point response to all the questions as listed below.
Point 1: I think the first paragraph in “Introduction” is a bit too long. I suggest breaking it down into 2-3 smaller paragraphs according to different main ideas (for example, one paragraph on GA biosynthesis and another paragraph on rice genes involved in this process).
Response 1: nice suggestion! We have divided the first paragraph of introduction into several small paragraphs. One main idea is about the GA biosynthesis, another is about the rice genes involved in GA biosynthesis.
Point 2: In the second paragraph in “Introduction”, could you briefly add why it is important to functionally characterize the role of OsABF1 on plant height and seed germination, probably by emphasizing the importance of those traits in agriculture? What about moving line 258-264 somewhere into “Introduction” instead of the “Discussion” section?
Response 2: Thank for your comments, which have been revised according to the suggestions.
Point 3: In line 69, the full name of bZIP should appear earlier when the term is used first in line 66.The first sentence in the result section (line 96) should be removed.
Response: Thank for your suggestion! We have corrected it.
Point 4: In Fig. S1 a and b, the genic locations of the CRISPR/Cas9 induced mutations should be indicated along with their protein-level effects. Which exon is affected by each mutation? Are they all frameshifts?
Response 4: Thank for your comments! As a matter of fact, after several generations’ sequencing detection, we only got two homozygous single mutant lines: abf1-1 and abf1-2. abf1-1 affected the first exon of OsABF1 CDS and caused frameshift. abf1-2 also affected the first exon of OsABF1 CDS and caused the T mutated into G, which changed the amino acid sequence. Meanwhile, we got two homozygous double mutant lines: abf1bzip40-1 and abf1bzip40-2. abf1bzip40-1 affected the first exon of OsABF1 and bZIP40 CDS respectively, and caused both of them frameshifts. abf1bzip40-2 affected the first exon of OsABF1 and bZIP40 CDS, respectively. For the abf1, there were “CTC” deletions in the first exon, which caused one amino acid deletion. For the bzip40, there was a “T” insertion into the first exon of bZIP40 CDS and caused the reading frame shift. The above of which (Line 128, Page 3) have been corrected.
Point 5: I am not convinced by line 130-132, which states OsABF1 might be functionally redundant with bZIP40 based on the observation that both abf1 single and abf1bizip40 double mutants showed no phenotypic effects on plant height and germination. Could you elaborate this more?
Response 5: Thank you for your attention. As we know that bZIP (Basic leucine zipper) transcription factors, as a highly conserved type of protein, are widely distributed in eukaryotes. There are about 110 and 89 bZIP transcription factors in Arabidopsis and rice, respectively. The differences in structural feature divided bZIPs into 10 subfamilies with diverse functions. As bZIP transcription factors are very conserved in diverse functions of plant growth and development, so may be knock out one or several members of bZIP transcription factors, the other members of bZIP transcription factors could make up the function of the bZIP transcription factors which were knocked out. So both the abf1 single and abf1bizip40 double mutants showed no obvious phenotypic effects on plant height and germination.
Point 6: Please revise the orders of supplemental figures so they appear in the text in alphanumerical order. Currently, Fig. S1c appears first followed by Fig. S2h, and then Fig. S1 a-b. This order is somewhat confusing for readers.
Response 6: I’m sorry to confuse you! The orders of supplemental figures have been modified.
Point 7: In line 157-160, indicate what “treatment” you are describing about.
Response 7: Thank for your comments! The line 157-160 “Hours After Treatment” was modified to “Hours After GA Treatment”.
Point 8: In Fig. 3c, the legend mentions that there are 10 bars (line 190) but only six bars are presented. The location of the G-box element mentioned in line 168 should be also included in Fig. 3c.
Response 8: Thanks for your comments! We have corrected according to your advice.
Reviewer 2 Report
In this manuscript, authors have shown that OsABF1 negatively regulates GA biosynthesis by repressing SD1 expression and affects plant development. Moreover, they have also shown that OsABF1 protein also interacts with polycomb repression complex component OsEMF2b by recruiting RRC2-mediated H3K27me3 deposition on the SD1 promoter. In my understanding, the data is quite interesting and presented well and the manuscript is written nicely. However, I have few comments/queries than can be addressed before acceptance.
- How many G boxes are present in the promoter regions? The data suggest that OsABF1 protein directly binds to SD1 promoter but it not clear whether OsABF1 specifically binds to the G box of SD1 promoters. For this, EMSA with mutated G box can be performed. Include EMSA probe sequence or G box(s) sequence/location within the P3 region in the figure 3, preferably right above EMSA. From the data, it seems that OsABF1 is sufficient to repress the SD1 expression.
- However, OsABF1 also has different mechanism of SD1 regulation. OsABF1 interacts with OsEMF2b and represses SD1 expression. Moreover, the data suggests that OsEMF2b is also a repressor of SD1 (figure 5f). Fig 3a, 5d and 5e suggest that OsABF1 and OsEMF2b inhibit the expression of each other. Under what kind of conditions, OsABF1 interacts with OsEMF2b in plants because both are antagonistic to each other? This should be discussed in more detail. Similarly, what could be the mechanism of OsABF1-mediated direct regulation or OsABF1 mediated indirect regulation by interaction with OsEMF2b to regulate GA biosynthesis and plant development.
- Page 5, line 159: GA should be mentioned.
- What age and plant tissue was used to check the expression of genes by qRT-PCR. Please mention this.
Author Response
Response to Reviewer 2 Comments
In this manuscript, authors have shown that OsABF1 negatively regulates GA biosynthesis by repressing SD1 expression and affects plant development. Moreover, they have also shown that OsABF1 protein also interacts with polycomb repression complex component OsEMF2b by recruiting RRC2-mediated H3K27me3 deposition on the SD1 promoter. In my understanding, the data is quite interesting and presented well and the manuscript is written nicely. However, I have few comments/queries than can be addressed before acceptance.
Response: Thank you for your appreciation! We fully considered the suggestive comments and made a point-to-point response to all the questions as listed below.
Point 1: How many G boxes are present in the promoter regions? The data suggest that OsABF1 protein directly binds to SD1 promoter but it not clear whether OsABF1 specifically binds to the G box of SD1 promoters. For this, EMSA with mutated G box can be performed. Include EMSA probe sequence or G box(s) sequence/location within the P3 region in the figure 3, preferably right above EMSA. From the data, it seems that OsABF1 is sufficient to repress the SD1 expression.
Response 1: Thank you for your attention! There are two G boxes present in the promoter of SD1, and both of them are located in the P3 region. As is know that OsABF1 is a member of bZIPs family, previous studies showed that bZIPs transcription factors are inclined to directly binding to the conserved G-box cis elements (CACGTG) in the promoter to activate or repress the gene transcription (EMBO. 1990, 9:1727-1735; New Phytologist 2020, 228: 1336–1353). In this study, we performed EMSA assay with competitive probes, the result showed that the GST-OsABF1 protein could directly binding to the SD1a probe (containing two G-boxes) in vitro. Meanwhile, when added the unlabeled competitive probes, which could effectively alleviated the biotin probe bound by GST-OsABF1 (Fig.3d). For these reasons above, we have the reasons to convince that the OsABF1 could specifically binds to the G box of the SD1 promoter. The EMSA probe sequences of the G-box have been added in the Fig 3d.
Point 2: However, OsABF1 also has different mechanism of SD1 regulation. OsABF1 interacts with OsEMF2b and represses SD1 expression. Moreover, the data suggests that OsEMF2b is also a repressor of SD1 (figure 5f). Fig 3a, 5d and 5e suggest that OsABF1 and OsEMF2b inhibit the expression of each other. Under what kind of conditions, OsABF1 interacts with OsEMF2b in plants because both are antagonistic to each other? This should be discussed in more detail. Similarly, what could be the mechanism of OsABF1-mediated direct regulation or OsABF1 mediated indirect regulation by interaction with OsEMF2b to regulate GA biosynthesis and plant development.
Response 2: Thank for your comments! Our research results showed that OsABF1 was able to physically interact with PRC2 component OsEMF2b in vitro and vivo (Fig. 4a-c).And there might be a regulation mechanism that over-expression of OsABF1 could repress the expression of SD1. At the meantime, enrichment of OsEMF2b would mediate more H3K27me3 in the promoter of SD1, which also repress its expression. Further, in plant, there must be a regulation mechanism of maintaining the GA homeostasis, namely the moderate and stable expression of SD1. So when OsABF1 was over-expressed, then the expression of OsEMF2b was reduced (Fig 5 e), whereas when OsEMF2b mutant to emf2b, then the expression of OsABF1 was increased significantly to repress the expression of SD1, and then maintain the moderate and stable expression and the GA homeostasis in plant, which we have added in the discussion section.
Point 3: Page 5, line 159: GA should be mentioned.
Response 3: We have modified the line 159 and added the GA in the sentence.
Point 4: What age and plant tissue was used to check the expression of genes by qRT-PCR. Please mention this.
Response 4: Thanks for your nice suggestions! The leaves of the tillering stage of the experiment materials were used to check the expression of genes by qRT-PCR, which were added in the manuscript.